

# Coordinative control of G2/M phase of the cell cycle by non-coding RNAs in hepatocellular carcinoma

Jun Shi[1,*], Guangqiang Ye[2,*], Guoliang Zhao[2], Xuedong Wang[1], Chunhui Ye[2], Keooudone Thammavong[2], Jing Xu[2] and Jiahong Dong[1]

[1] Department of Hepatobiliary and Pancreas Surgery, Beijing Tsinghua Changguang Hospital, School of Clinical Medicine, Tsinghua University, Beijing, China
[2] The First Affiliated Hospital of Guangxi Medical University, Guangxi Medical University, Nanning, China
* These authors contributed equally to this work.

Corresponding authors
Jing Xu, xujing@gxmu.edu.cn
Jiahong Dong,
dongjiahong@mail.tsinghua.edu.cn

## ABSTRACT

**Objective:** To investigate the interaction of non-coding RNAs (ncRNAs) in hepatocellular carcinoma.

**Methods:** We compared the ncRNAs and mRNAs expression profiles of hepatocellular carcinoma and adjacent tissue by microarray and RT-PCR. The relationship between different ncRNAs and mRNA was analyzed using bioinformatics tools. A regulatory model of ncRNAs in hepatocellular carcinoma cells was developed.

**Results:** A total of 1,704 differentially expressed lncRNAs, 57 miRNAs, and 2,093 mRNAs were identified by microarray analyses. There is a co-expression relationship between two ncRNAs (miRNA-125b-2-3p and lncRNA P26302). Bioinformatics analysis demonstrated cyclin-dependent kinases 1 and CyclinA2 as potential targets of miR-125b-2-3p and Polo-like kinase 1 as potential target of lncRNAP26302. All three gene are important components in the G2/M phase of cell cycle. Subsequently real-time polymerase chain reaction (PCR) studies confirmed these microarray results.

**Conclusion:** MiR-125b-2-3p and lncRNAP26302 may affect the G2/M phase of the cell cycle through the regulation of their respective target genes. This study shows a role of ncRNAs in pathogenesis of hepatocellular carcinoma at molecular level, providing a basis for the future investigation aiming at early diagnosis and novel treatment of hepatocellular carcinoma.

## INTRODUCTION

Hepatocellular carcinoma (HCC) is the most common primary liver cancer, accounting for 85–90% of all primary livery cancer cases. Among the world's most common malignancies, the incidence of HCC is ranked sixth, and its mortality rate is ranked third (*Torre et al., 2015*). The high mortality rate is mainly due to the strong invasive and metastatic capacity of HCC cells. Upon diagnosis, HCC is often at an advanced stage (*Wang et al., 2013*). Therefore, to improve the prognosis and treatment

of liver cancer, investigators have been working to understand the cellular and molecular biological mechanisms leading to the malignant transformation of hepatocytes. Early diagnosis is the key to effective treatment of liver cancer. It is important to find effective early diagnosis markers of liver cancer. In recent years, a large number of studies have suggested that non-coding RNAs (ncRNA) are related to the occurrence of tumors, providing a new research direction for the diagnosis and treatment of liver cancer (*Mattick & Makunin, 2006*).

As more and more ncRNAs are studied, it is notable that different ncRNAs have different regulatory capabilities in the development of liver cancer. MicroRNA (miRNA) and long ncRNA (lncRNA) are the two most representative forms. For example, miR-122 is a miRNA specifically expressed in the liver and thus overexpressed in HCC cell lines HepG2 and HepB3. This miRNA promotes apoptosis and inhibits proliferation of hepatoma cells in biological processes (*Datta et al., 2008*). In addition, miR-122 may also target the activity of p53 through the cyclin G1 gene to affect the sensitivity of HCC to doxorubicin (*Hsu et al., 2012*). In a study by *Tsang et al. (2015)*, the lncRNA HOTTIP was identified as the most significantly up-regulated lncRNA in human HCCs, even in an early stage of HCC formation. Functionally, knock-down of HOTTIP attenuated HCC cell proliferation in vitro and markedly abrogated tumorigenicity in vivo. In addition, knock-down of HOTTIP also inhibited the migratory ability of HCC cells and significantly abrogated lung metastasis in an orthotopic implantation model in nude mice. Furthermore, they identified miR-125b as a post-transcriptional regulator of HOTTIP. Ectopic expression of miR-125b reduced HOTTIP-coupled luciferase activity and suppressed the endogenous level of HOTTIP. lncRNA HULC is highly expressed in HCC, while it has a binding site of miR-372. Therefore, HULC can competitively bind miRNAs, making miRNAs lose their ability to target genes, thereby reducing the tumor suppressor effect of miR-372 (*Wang et al., 2010*). Researchers have termed the ncRNAs that competitively bind to miRNA as competitive endogenous RNA (ceRNA) (*Salmena et al., 2011*). This concept has enriched the central dogma of molecular biology and provided a useful tool for analyzing the molecular mechanisms of tumors.

During the study of a single ncRNA, it was found that the effect of certain ncRNAs on tumorigenesis is accompanied by corresponding changes in other types of ncRNA. In this regard, we hypothesize that ncRNAs among different species may play a regulatory role in the development of tumors by coordinatively acting on the same target gene or upstream and downstream genes on the same signaling pathway. In order to explore whether other regulatory mechanisms of lncRNAs and miRNAs are involved in HCC pathogenesis, we compared the expression levels of lncRNA, miRNA, and mRNA between HCC tissues and adjacent normal tissues. Bioinformatics prediction, correlation analysis, and pathway evaluation were used to identify a potentially important ncRNA and its target genes.

## MATERIALS AND METHODS

### Materials

In this study, 12 pairs of primary HCC tissues and their adjacent tissues collected from two cm away from the cancer tissue were selected (*Chen et al., 2016*). All specimens were

obtained from the First Affiliated Hospital of Guangxi Medical University. The tissue samples were processed within 30 min after the tumor was isolated. They were immediately frozen in liquid nitrogen and stored at −80 °C. The patients from which the specimens were sourced were all undergoing a first operation on primary disease, and none had received radio-chemotherapy. Histopathology confirmed the diagnosis of HCC after surgery. The use of all specimens was reviewed by the Ethics Committee of the First Affiliated Hospital of Guangxi Medical University, and patients and their families provided informed consent for participation in scientific research. The use of all specimens was reviewed by the Ethics Committee of the First Affiliated Hospital of Guangxi Medical University. Ethical Application Ref: 伦审2013-KY-国基-085.

## RNA preparation

Total RNA in tissue blocks was extracted using Trizol (Invitrogen, Carlsbad, CA, USA). Total RNA was further purified using the NucleoSpin® RNA clean-up kit (740.948.250) and the mirVanaTM miRNA Isolation Kit (AM1561). The prepared total RNA will be used for the next experiment.

## miRNA microarray

We performed microarray analysis of miRNAs using Affymetrix GeneChip miRNA arrays (Santa Clara, CA, USA). In short, polyA polymerase uses the Genisphere FlashTag HSR kit to label one mg of total RNA from the tissue. This was followed by hybridization of RNA with Affymetrix miRNA arrays. After hybridization, the standard Affymetrix ribbons were stained, washed, scanned, and transformed with AGCC software (Affymetrix® GeneChip® command console® software) and the GeneChip® Scanner 3000. Using .CEL files as source files, we use the Expression Console software provided by Affymetrix to perform row data preprocessing: including RNA normalization, if the probe signal was significantly higher than the background signal, and integration of the probe signal into a probe set signal. Three miRNA microarray tests were performed on each sample.

## lncRNA and mRNA microarray

The Jingxin® Human lncRNA + mRNA Expression Microarray (CapitalBio, Beijing, China) screens lncRNA and mRNA at the same time. According to the manufacturer's instructions, five μL total RNA extracted from each sample was used to synthesize double-stranded complementary DNA (cDNA). Double-stranded cDNA was labeled and hybridized to the Jingxin® Human lncRNA + mRNA Expression Microarray. The volume of the cDNA product obtained after reverse transcription and purification was concentrated to 14 μL, and cy3-dCTP/cy5-dCTP was added and placed in a PCR machine (reaction at 37 °C for 1.5 h, reaction at 70 °C for 5 min, retention at 4 °C). Then, the cDNA was purified using the Nucleospin® Extract II (MN, Cat. No. 740609.250) kit, and the fluorescently labeled product was subjected to fluorescence incorporation and nucleic acid quantification using an ultraviolet spectrophotometer. Next, 100 μL of the hybridization solution was applied to the hybrid coverslip, and the hybridization

cassette was mounted on the rotor of the Hybridization Oven G2545A and hybridized at 45 °C for 12 h. The cleaned chip was scanned with an Agilent G2565CA Microarray Scanner to obtain a hybrid picture. Hybrid images were analyzed and data extracted using Agilent Feature Extraction (v10.7) software. Then we used Agilent GeneSpring software to normalize and analyze the data.

## RT-PCR detection

Total RNA was isolated from samples using the Trizol reagent, and then cDNA was synthesized using viral polymerase and random primers. The cDNA PCR products were amplified, and the target gene expression levels were normalized to 18s mRNA, an internal quantity control. Experiments were performed in triplicate.

## Statistical analysis

We analyzed the differentially expressed genes in the preprocessed data. Three or more biologically replicated data points for each sample were analyzed using the SAM (significance analysis of microarray) R package (*Tusher, Tibshirani & Chu, 2001*). The screening criteria for differential genes were: $q$-value $\leq 5\%$ and fold change $\geq 2$ or $\leq 0.5$ (*Clarke et al., 2008*; *Yang et al., 2005*). The genes identified as differentially expressed were subjected to unsupervised hierarchical clustering (Cluster 3.0) and TreeView analysis (Stanford University, Stanford, CA, USA). Moreover, gene ontology (GO) enrichment analysis was carried out for differentially expressed mRNAs. The array data presented here is accessible through GEO series accession number GSE115019.

## RESULTS

### Differentially expressed miRNAs, mRNAs, and lncRNAs in HCC

An Affymetrix miRNA microarray was used to establish the miRNA expression profiles of 12 pairs of human HCC tissues and their adjacent tissues. Based on a fold change of $\geq 2$ or $\leq -2$ with $P < 0.05$ for the $t$-test as the screening criteria for differential genes, 57 differentially expressed miRNAs were identified, including nine that were up-regulated and 48 that were down-regulated (Fig. 1A). The same samples were used for gene expression profiling using the Jingxin® human lncRNA + mRNA V4.0 chip. Compared to the adjacent tissue, HCC tissues showed significant differences in the gene expression levels of 2,093 mRNAs, among which 635 were up-regulated and 1,458 were down-regulated (Fig. 1B). A total of 1,704 lncRNAs showed significantly different expression levels in HCC tissues, of which 607 were up-regulated and 1,097 were reduced (Fig. 1C). The GO enrichment analysis results showed that differential mRNA sets were enriched in mitotic cell cycle, DNA packaging, cell cycle phase, etc. (Fig. 2).

### Co-expression analysis constructs and signaling pathways

We performed a co-expression analysis of lncRNAs and miRNAs in order to select the co-expressed genes based on an absolute value of correlation coefficient >0.7 and $P < 0.05$. The lncRNAs and miRNAs with a co-expression relationship were used to predict

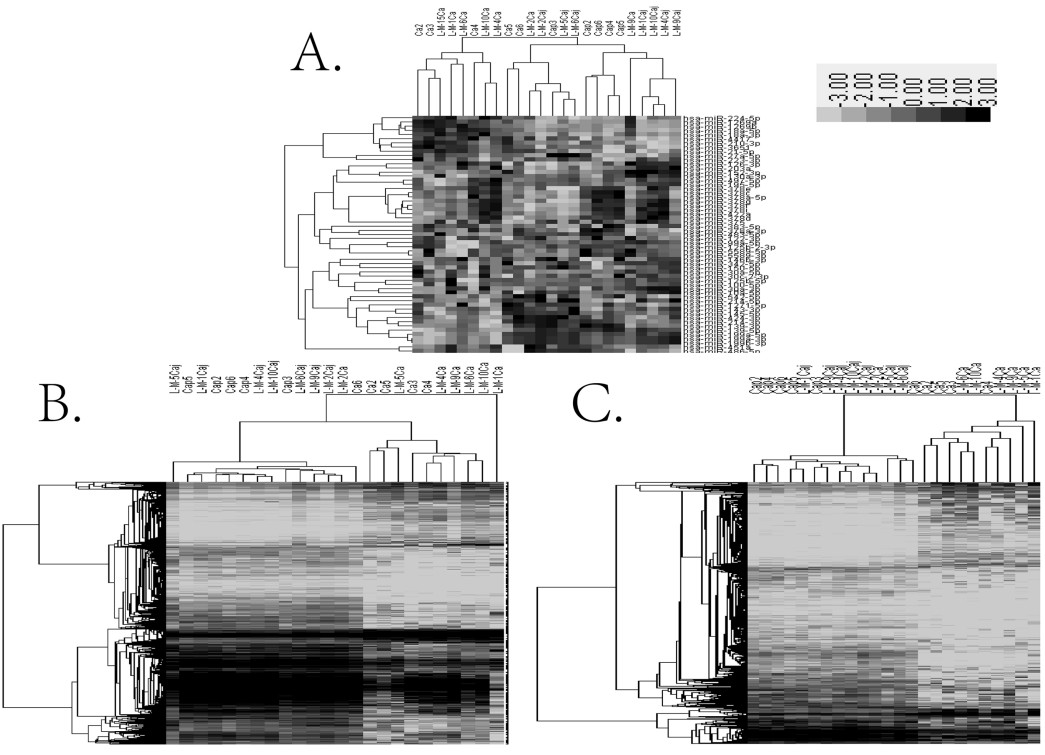

**Figure 1  The result of cluster analysis.** (A) miRNA cluster analysis; (B) lncRNA cluster analysis; (C) mRNA cluster analysis.                           

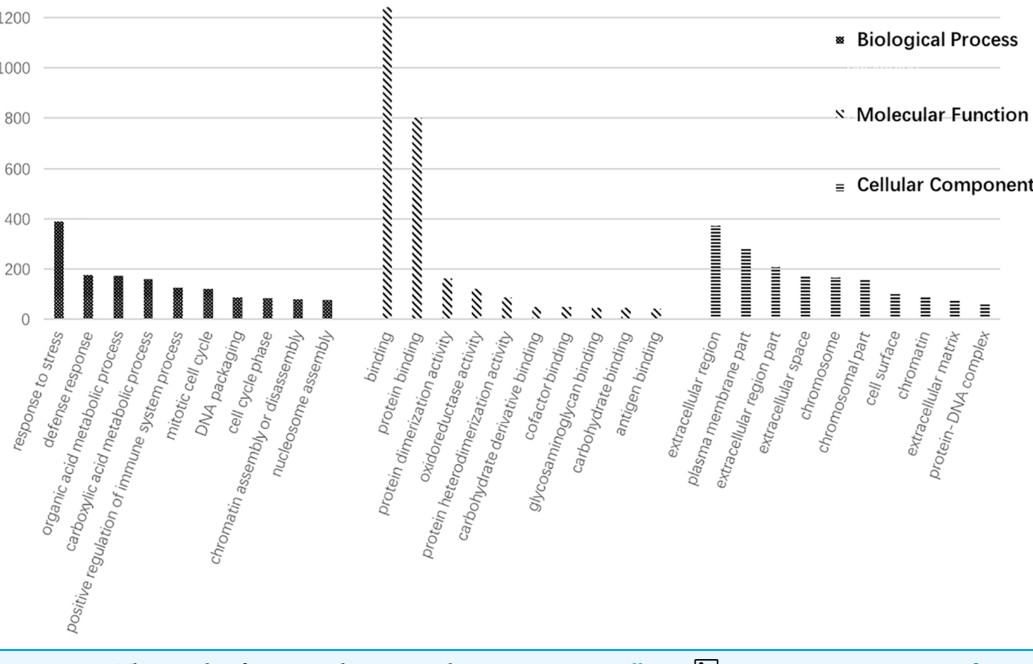

**Figure 2  The result of Go enrichment analysis.**                           

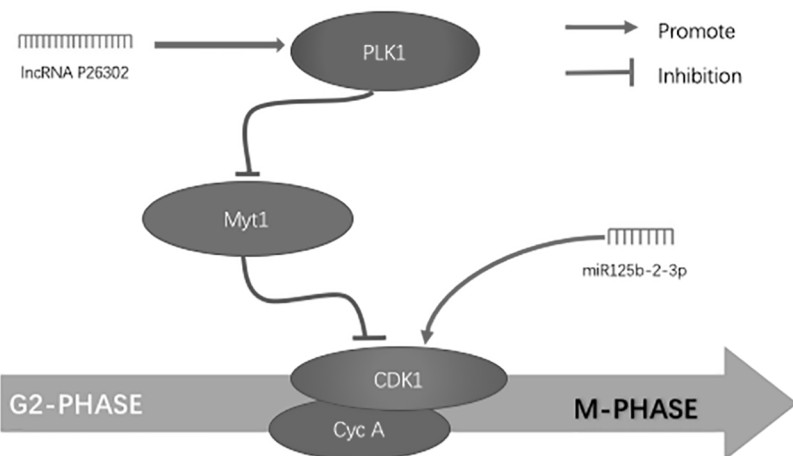

**Figure 3 Simultaneous control of non-coding RNA on the G2/M phase of the cell cycle in hepatocellular carcinoma cells.**

their target genes, which were cross-checked with genes identified by mRNA microarray analysis. We identified 20 co-expressed lncRNAs and miRNAs, of which only lncRNA P26302 had a fold difference >10×, so we chose miR-125-2-3p co-expressed with lncRNAP26302 (correlation coefficient = −0.87, $P = 0.0001$) for further study. Meanwhile, three differentially expressed mRNAs that were target genes of miR-125b-2-3p or lncRNA P26302 were identified by mRNA microarray. The target genes for miRNA-125b-2-3p were cyclin-dependent kinases 1 (CDK1) and cyclin A2. Polo-like kinase 1 (PLK1) is a target gene of lncRNA P26302. Through KEGG pathway analysis of target genes, we found that the three target genes co-exist in the G2/M phase of the cell cycle pathway, and they are close in order within the pathway. Thus, we propose the signal pathways shown in Fig. 3.

## RT-PCR verification of differential gene expression

To verify the differential expression of genes identified by microarray screening, we performed RT-PCR analyses. The results showed that the expression of miR125b-2-3p was significantly lower in HCC tissues than in adjacent tissues ($P_{miR125b-2-3p} = 0.0052$), while the expression levels of lncRNAp26302, CDK1, cyclin A2, and PLK1 were significantly higher in HCC tissues than in adjacent tissues ($P_{P26302} = 0.0255$; $P_{CCNA\ 2} = 0.028$; $P_{CDK\ 1} = 0.0171$; $P_{plk\ 1} = 0.0267$). The differential gene expression detected in HCC and adjacent tissues was consistent with the results of the gene chip screening (Fig. 4).

## DISCUSSION

Non-coding RNAs such as lncRNAs and miRNAs have been proven to participate in the regulation of gene expression through competition for endogenous RNA networks with mRNA (*Kartha & Subbaya, 2014*). The interaction between these two ncRNAs plays an important role in tumor development (*Jiang et al., 2017*; *Wu et al., 2015*). At present, the specific molecular mechanism of the interaction between lncRNA and miRNA is speculated to be one of two possibilities: (1) because lncRNAs and mRNAs have a

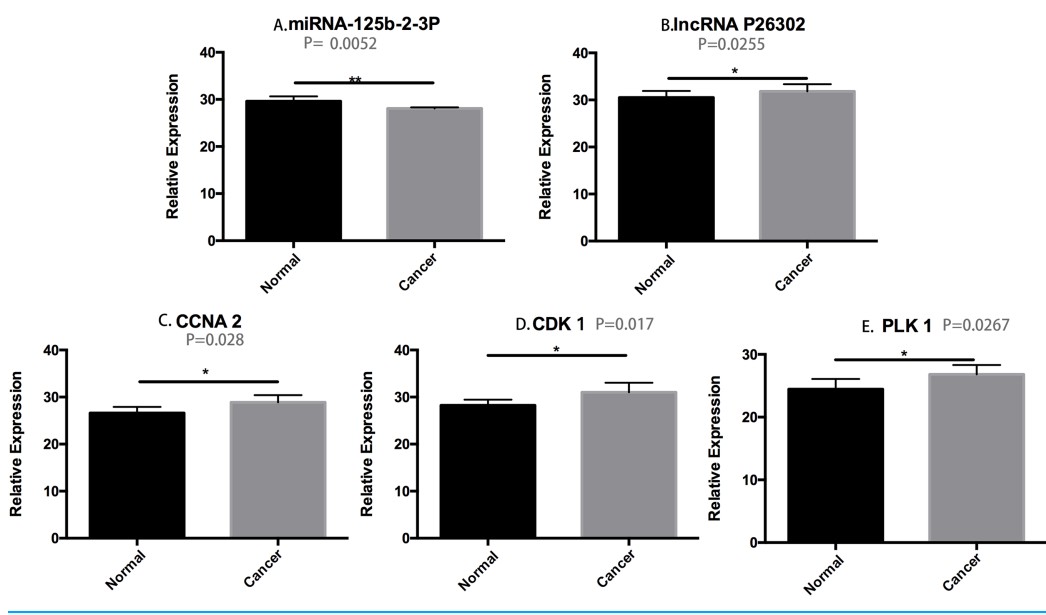

**Figure 4 The results of differential gene RT-PCR.** (A) $P_{\text{mir125b-2-3p}}$ = 0.0052; (B) $P_{\text{P26302}}$ = 0.0255; (C) $P_{\text{CCNA 2}}$ = 0.028; (D) $P_{\text{CDK 1}}$ = 0.0171; (E) $P_{\text{plk 1}}$ = 0.0267.

similar structure, a miRNA can specifically bind to its 3′ UTR and down-regulate lncRNA expression through a mechanism similar to that of mRNA regulation (*Shi et al., 2013*); or (2) studies have confirmed that lncRNAs can competitively target miRNAs to inhibit the expression of miRNAs, thereby reducing their inhibitory effect on target genes (*Salmena et al., 2011*).

During the G2 to M phase transition of the cell cycle, CDKs represent a set of Ser/Thr kinase systems that correspond to cell cycle progression. Various CDKs are alternately activated along the cell cycle, and phosphorylation of the corresponding substrate allows the cell cycle to proceed in an orderly manner. Cyclin A2 is a cyclin that binds to CDK1 to form a CycA/CDK1 complex (*Santamaría et al., 2007*). The CycA/CDK1 complex is the main damage monitoring mechanism in S phase. The withdrawal of the cell cycle from mitosis requires CDK1 inactivation, and the most important mechanism of CDK1 inactivation is the hydrolysis of mitotic cyclins. In higher eukaryotes, this involves the continuous destruction of type A and type B cyclins, which results in successive inactivation of the CycA/CDK1 complex and the CycB/CDK1 complex. When DNA damage occurs, Cyclin A is first destroyed, resulting in the inactivation of CycA/CDK1 necessary for the G2 to M transition (*Kaspar et al., 2001*). Finally, the cell cycle is stopped in the G2 phase. PLK1 represents a class of highly conserved serine/threonine protein kinases expressed in eukaryotes (*Takaki et al., 2008*). It has been proven in many studies that Plk1 is a key gene in cell cycle regulation and is regulated by phosphorylation and protein degradation (*Barr, Sillé & Nigg, 2004*) (*Catherine & Jonathon, 2004*). Current studies have found that PLK1 is able to restart mitosis by acting on the CycA/CDK1 complex, from the G2 phase to the M phase. In the G2 phase, the cytokine CycA/CDK1 complex is inactivated due to the phosphorylation of the residues on the

adenosine triphosphate-binding domain of Cdk1, while Myt1 kinase phosphorylates the threonine residue Tyr15. This stops entrance into the mitosis phase before DNA replication is completed and destroys genome integrity (*Takizawa & Morgan, 2000*). Plk1 is able to phosphorylate and inhibit the activation of Myt1, resulting in dephosphorylation of CDK1, and activation of CycA/CDK1 complex initiates mitosis (*Nakajima et al., 2003*).

To further explore the relationship among the three groups, we performed co-expression analysis of the differentially expressed miRNAs and lncRNAs and analyzed the KEGG enrichment pathways of their respective target genes. We found that miRNA125b-2-3p and lncRNAP26302 expression levels correlated in HCC. Moreover, the target genes CDK1 and cyclin A2 of miR-125b-2-3p are in the same gene pathway (the G2/M phase of the cell cycle) and of the target gene PLK1 of lncRNAP26302. In the normal cell cycle, CycA/CDK1 is the major damage regulation mechanism of S phase (*Katsuno et al., 2009*), and cyclin A2 in CycA can promote the synthesis of DNA in S phase, thus promoting the transition of cells from the G2/M phase into M phase (*Arsic et al., 2012*). However, when DNA damage occurs, cell cycle regulation is influenced by the DNA damage repair mechanism, which inhibits the expression of CycA/CDK1 through the Myt1 gene (*Varadarajan et al., 2016*), thus allowing the cell cycle to stay in the S phase for cell repair (*Chapman, Taylor & Boulton, 2012*; *Inger & Gent, 2012*). In hepatoma cells, the target gene PLK1 of lncRNAP26302 was overexpressed, inhibiting the expression of Myt1 and reducing the inhibitory effect of Myt1 on CycA/CDK1. As a result, when DNA damage or mutation occurs in the nuclei of tumor cells, the chromosomes can bypass the DNA damage detection point, and these cells continue to proliferate and divide, thereby promoting the formation of HCC (*Perdiguero & Nebreda, 2004*). In addition, chip detection revealed that low expression of miR-125b-2-3p leads to an increase in the expression of the target gene cyclin A2, further promoting the formation of CycA/CDK1. By differential gene correlation analysis, a reverse correlation was found between miR-125b-2-3p and lncRNAP26302, which indicates that the decreased expression of miR-125b-2-3p in HCC may cause overexpression of lncRNAp26302. Therefore, miR-125b-2-3p and lncRNAP26302 may have a coordinative effect on this pathway, together promoting the expression of CycA/CDK1, so as to coordinatively promote the formation of liver cancer.

We also performed RT-PCR validation of the differential expression of the genes regulated by the identified lncRNAs. Among them, the RT-PCR results for hs-mir125b-2-3p, lncRNAp26302(uc003lzi.3), CDK1, cyclin A2, and PLK1 were consistent with their expression in the G2/M phase of the cell cycle. Therefore, we believe that miRNA 125b-2-3p and lncRNA p26302 have a coordinative regulatory effect on the G2/M phase in HCC.

## CONCLUSIONS

In summary, a large number ncRNAs are differentially expressed between HCC tissue and adjacent normal tissue. From the expression profiling study of these genes, we propose that mir-125b-2-3p and lncRNA p26302 affect the G2/M phase of the cell cycle through a coordinative regulation of their respective target genes. The gene expression results

of this study expand our understanding of the gene expression mechanism of HCC, providing support for the ceRNA hypothesis and a more complete understanding of the role of ncRNAs in the occurrence and development of HCC and the regulatory loop. In the future, functional cell cycle analysis is needed to demonstrate the role of miR-125b-2-3p and lncRNAP26302 target genes in the cell cycle G2/M phase.

### Funding
This work was supported by the National Natural Science Foundation of China (No. 81360372). The funders had no role in study design, data collection and analysis, decision to publish, or preparation of the manuscript.

### Grant Disclosures
The following grant information was disclosed by the authors:
National Natural Science Foundation of China: 81360372.

### Competing Interests
The authors declare that they have no competing interests.

### Author Contributions
- Jun Shi conceived and designed the experiments, performed the experiments, analyzed the data, contributed reagents/materials/analysis tools, prepared figures and/or tables, authored or reviewed drafts of the paper, approved the final draft.
- Guangqiang Ye conceived and designed the experiments, performed the experiments, analyzed the data, contributed reagents/materials/analysis tools, prepared figures and/or tables, authored or reviewed drafts of the paper, approved the final draft.
- Guoliang Zhao analyzed the data, contributed reagents/materials/analysis tools, approved the final draft.
- Xuedong Wang analyzed the data, contributed reagents/materials/analysis tools, approved the final draft.
- Chunhui Ye prepared figures and/or tables, approved the final draft.
- Keooudone Thammavong prepared figures and/or tables, approved the final draft.
- Jing Xu conceived and designed the experiments, authored or reviewed drafts of the paper, approved the final draft.
- Jiahong Dong conceived and designed the experiments, authored or reviewed drafts of the paper, approved the final draft.

### Human Ethics
The following information was supplied relating to ethical approvals (i.e., approving body and any reference numbers):

The use of all specimens was reviewed by the Ethics Committee of the First Affiliated Hospital of Guangxi Medical University. Ethical Application Ref: 伦审2013-KY-国基-085.

## Data Availability

The array data discussed here is accessible through GEO series accession number GSE115019.

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
