# Peer review of "Coordinative control of G2/M phase of the cell cycle by non-coding RNAs in hepatocellular carcinoma"

_PeerJ, doi:10.7717/peerj.5787_

## Round 0.1 · original submission · Major Revisions

Please provide a point-to-point response to all comments from the reviewers.

Careful review of the grammar and formatting in the text is recommended.

Reviewer 1 ·

Basic reporting

This manuscript reports some interesting data obtained by microarray analysis of human liver cancer tissues and matched adjacent normal tissues. Further bioinformatics analysis was also performed to identify some signaling pathway or important genes regulated by differentially expressed non-coding RNAs.

Experimental design

The experimental design in the microarray parts was good.

Validity of the findings

The authors claimed that "MiR-125b-2-3p and lncRNAP26302 may affect the G2/M phase of the cell cycle through the regulation of their respective target genes", however, no cell cycle analysis was carried out. The authors need to either revise their conclusion or perform experiments to demonstrate the role of MiR-125b-2-3p and lncRNAP26302 target gene in cell cycle G2/M phase.

Additional comments

The manuscript needs further English editing.

Reviewer 2 ·

Basic reporting

The article uses clear English to describe the effect of MiR-125b-2-3p and IncRNAP26302 on the G2/M phase of the cell cycle through the regulation of CHEK1 and CCNA2 and PLK1.

Tables and figures are appropriate. Results support their hypotheses. There are sufficient references to support their introduction and discussion.
Introduction need more detailed information (e.g. why non-coding RNA research is so important; review more current research in HCC (e.g. HOTTIP and Mir-125b); indicate how your research fills an identified knowledge gap; state the value of your research).

Simply repeating the results (like line 196-209) is not necessary in Discussion section. In addition, it would be better if you can compare and contrast your findings with those of other published results in your discussion. You also need to indicate how your research work is important in the early diagnosis, targeted therapy and prognosis of HCC in the discussion.

Several spelling mistakes are noted including line 45 understanidng (understanding) and line 56 blinding (binding)

Some abbreviations need to be followed by the full name in parentheses, e.g. IncRNA, HULC, CHEK, CCNA, and PLK.

Experimental design

Crosstalk between long noncoding RNAs and MicroRNAs in disease is a hot research field in recent years. Only a few studies involving interaction between IncRNA and MicroRNAs in hepatocellular carcinoma have been published (Liver Int. 2015;35:1597–1606 and Nucleic Acids Res. 2010;38:5366–5383).

This experiment used miRNA, IncRNA and mRNA microarray to identify the novel correlation between MiR-125b-2-3p and IncRNAP26302 in HCC and demonstrate the possible G2/M gene targets, which has not been reported and is a totally novel finding.

The methods described provide sufficient details and information to replicate.

Validity of the findings

The data of this article is totally novel, solid and statically sound and the conclusion are well stated, linked to original research question.

Regarding result1, it would be better if authors could summarize some common features of top significantly dysregulated genes (e.g. they also involve the specific pathway)

Regarding result2, how many co-expressed IncRNAs and miRNAs have been identified? Why do you only report miR125b-2-3p and IncRNA P26302? Is it due to highest correlation coefficient and/or lowest P value?

Regarding result3, it would be better if you use one or two sentences to highlight the most important points.

Additional comments

Overall, this paper is well- designed. The data is novel and solid, and has not been reported.

However, more detailed information is needed in your introduction. Further discussion is suggested in order to support your novel findings and to show the unique value of your research.

Reviewer 3 ·

Basic reporting

Shi et al. had examined the hepatocellular carcinoma tissue and surrounding nourmal tissue of 12 patients and generated the profile of lncRNAs, miRNAs and mRNAs. They demonstrated that CHEK1 and CCNA2 are targets of miRNA-125b—2-3p; and PLK is targeted by lncRNp26302.
The authors are trying to conclude that miRNA-125b—2-3p and lncRNp26302 are thus regulating cell cycle at specifically G2/M phase.

Experimental design

This study is based on RNA screenings, not at the protein level. Thus whether the effect of miRNA-125b—2-3p and lncRNp26302 on the mRNA of CHEK1, CCNA2 and PLK can be carried on to translational products or cellular functionality are not clear. It is still too early to conclude any clinical relationship between these two RNAs and the proliferation or progression of hepatocellular carcinoma. Furthermore, the molecular function of miRNA-125b—2-3p and lncRNp26302 in regulating mRNAs of CHEK1, CCNA2 and PLK is also inconclusive without gain of function or loss of function experiments. Overall, more experiments are required to rule out the possibility of pure correlation, and to confirm the causal relationship.

Validity of the findings

1.Please make sure to check up the formatting of the manuscript. Some of the periods are not in correct form.
2.There is inconsistency in the miRNA number that are studied. In the abstract it is 81 while in the results it is 57.
3.The correlation analysis showed high score between miRNA-125b—2-3p and lncRNp26302, however the author should make it clearer that there is no direct molecular function between these two RNAs. Instead, they each have their own targets. so in other words the correlation, between the target mRNA expressions, is not due to any result of biological functionality between their upstream regulators.
4.The author should refer to the name of protein instead of the gene name, like CCNA2 in the Kegg pathway, which makes readers hard to track it and understand it.

Additional comments

The title use “simultaneous” to describe the regulation on cell cycle by miRNA-125b—2-3p and lncRNp26302, which is ambiguous. May be changed to “Coordinative control of G2/M phase of the cell cycle by non-coding RNAs in hepatocellular carcinoma”.

Annotated reviews are not available for download in order to protect the identity of reviewers who chose to remain anonymous.

---

## Round 0.2 · Minor Revisions

In addition to the reviewer 2's additional comment, please address the following comments which were missed in your revision. The lack of data on cell cycle phenotype is particularly concerning. Removal of these words in the title and softening the conclusion are recommended if no additional experiments are carried out.

R1,
The authors claimed that "MiR-125b-2-3p and lncRNAP26302 may affect the G2/M phase of the cell cycle through the regulation of their respective target genes", however, no cell cycle analysis was carried out. The authors need to either revise their conclusion or perform experiments to demonstrate the role of MiR-125b-2-3p and lncRNAP26302 target gene in cell cycle G2/M phase.

R2,
Regarding result1, it would be better if authors could summarize some common features of top significantly dysregulated genes (e.g. they also involve the specific pathway)

Regarding result3, it would be better if you use one or two sentences to highlight the most important points.

R3,
Overall, more experiments are required to rule out the possibility of pure correlation, and to confirm the causal relationship.

I also edited some grammatical issues in the abstract and the main text (see attached pdf file). Please use as you like.

Reviewer 1 ·

Basic reporting

This is a revised manuscript and my concerns have been addressed.

Experimental design

This is a revised manuscript and my concerns have been addressed.

Validity of the findings

This is a revised manuscript and my concerns have been addressed.

Additional comments

This is a revised manuscript and my concerns have been addressed.

Reviewer 2 ·

Basic reporting

as the first review

Experimental design

as the first review

Validity of the findings

as the first review

Additional comments

The authors did well responses for most concern and question, and modified the manuscript following all reviewers' comments. As the authors mentioned, it is a a prospective study.More results need be discussed in future. Currently, The authors need better explain and double-check the data from Fig4., It is the only evidence to verify the microarray findings.

Reviewer 3 ·

Basic reporting

no comment

Experimental design

This study is based on RNA screenings, not at the protein level. Thus whether the effect of miRNA-125b—2-3p and lncRNp26302 on the mRNA of CHEK1, CCNA2 and PLK can be carried on to translational products or cellular functionality are not clear. It is still too early to conclude any clinical relationship between these two RNAs and the proliferation or progression of hepatocellular carcinoma. Furthermore, the molecular function of miRNA-125b—2-3p and lncRNp26302 in regulating mRNAs of CHEK1, CCNA2 and PLK is also inconclusive without gain of function or loss of function experiments. Overall, more experiments are required to rule out the possibility of pure correlation, and to confirm the causal relationship.
It had been addressed.

Validity of the findings

no comment

Additional comments

1. Please make sure to check up the formatting of the manuscript. Some of the periods are not in correct form.
It had been addressed.
2. There is inconsistency in the miRNA number that are studied. In the abstract it is 81 while in the results it is 57.
It had been addressed.
3. The correlation analysis showed high score between miRNA-125b—2-3p and lncRNp26302, however the author should make it clearer that there is no direct molecular function between these two RNAs. Instead, they each have their own targets. so in other words the correlation, between the target mRNA expressions, is not due to any result of biological functionality between their upstream regulators.
It had been addressed.
4. The author should refer to the name of protein instead of the gene name, like CCNA2 in the Kegg pathway, which makes readers hard to track it and understand it.
It had been addressed.
5.The title use “simultaneous” to describe the regulation on cell cycle by miRNA-125b—2-3p and lncRNp26302, which is ambiguous. May be changed to “Coordinative control of G2/M phase of the cell cycle by non-coding RNAs in hepatocellular carcinoma”.
It had been addressed.

---

## Round 0.3 · Minor Revisions

Many thanks for submitting your interesting work to PeerJ!

---

## Round 0.4 · accepted · Accept

Congratulations! Many thanks for submitting your work to PeerJ!

#